# Are We What We Eat? Impact of Diet on the Gut–Brain Axis in Parkinson’s Disease

**DOI:** 10.3390/nu14020380

**Published:** 2022-01-17

**Authors:** Margherita Alfonsetti, Vanessa Castelli, Michele d’Angelo

**Affiliations:** Department of Life, Health and Environmental Sciences, University of L’Aquila, 67100 L’Aquila, Italy; margherita.alfonsetti@guest.univaq.it

**Keywords:** probiotics, prebiotics, synbiotics, Parkinson’s disease, neurodegeneration, α-synuclein, brain, gut, microbiota, diet

## Abstract

Parkinson’s disease is characterized by motor and non-motor symptoms, such as defects in the gut function, which may occur before the motor symptoms. To date, there are therapies that can improve these symptoms, but there is no cure to avoid the development or exacerbation of this disorder. Dysbiosis of gut microbiota could have a crucial role in the gut–brain axis, which is a bidirectional communication between the central nervous system and the enteric nervous system. Diet can affect the microbiota composition, impacting gut–brain axis functionality. Gut microbiome restoration through probiotics, prebiotics, synbiotics or other dietary means could have the potential to slow PD progression. In this review, we will discuss the influence of diet on the bidirectional communication between gut and brain, thus supporting the hypothesis that this disorder could begin in the gut. We also focus on how food-based therapies might then have an influence on PD and could ameliorate non-motor as well as motor symptoms.

## 1. Introduction

Parkinson’s disease (PD) is a progressive neurodegenerative disorder characterized by the loss of dopaminergic neurons in the midbrain, specifically in the substantia nigra pars compacta, associated with the formation of cytoplasmatic inclusions named Lewy bodies constituted of insoluble α-synuclein aggregates [1]. Nevertheless, PD pathology involves the degeneration of non-dopaminergic neurons as well. Interestingly, α-synuclein deposition is observed in several peripheral organs including the gastrointestinal (GI) system (submandibular gland, stomach, and bowels) raising the question of a possible role of the gut in PD pathogenesis [2].

PD is a complex disorder with multifactorial etiology: both environmental and genetic factors participate in a common set of pathways including mitochondrial alteration [3], ROS formation [4], protein aggregation, compromised autophagy and neuroinflammation [5,6].

PD clinical diagnosis is centered on the observation of both motor and non-motor symptoms. Rest tremor, bradykinesia, rigidity and loss of postural reflexes are the cardinal PD motor symptoms. Rather, secondary motor signs include dysarthria, glabellar reflexes, dysphagia, sialorrhea, micrographia, festination, shuffling gait, hypomimia and dystonia [7]. Although PD is considered a movement disorder, it is associated with a wide spectrum of non-motor features, such as anosmia, depression, sleep disorders, gastrointestinal dysfunction and low-grade mucosal inflammation in the enteric nervous system [8]. During later stages of the disorder, additional non motor features may appear that include autonomic dysfunction (orthostatic hypotension and urogenital dysfunction), pain and cognitive deficits [9].

To date, there are treatments that could help relief these symptoms but there is no cure to control the development and progression of this disease [10]. Pharmacological treatments include dopamine-based preparations such as levodopa, dopamine agonists and monoamine oxidase-B (MAO-B) inhibitors are usually administered as initial treatments [11]. Levodopa is the most commonly used drug, which controls some motor symptoms and counteracts dopaminergic cell loss by improving dopamine synthesis. However, this therapy has numerous adverse effects, it does not inhibit dopaminergic neurodegeneration and has no impact on non-motor symptoms [12]. Moreover, PD-associated gastrointestinal dysfunction contributes to levodopa effect fluctuations and oral treatment with levodopa needs optimal GI function to determine an ideal drug metabolism, indeed, it has been demonstrated that levodopa leads to delayed gastric emptying in healthy patients [13] and aggravates GI symptoms in PD patients [14]. In addition, further studies found that levodopa responsivity is directly correlated with the severity of α-synuclein accumulation in the enteric nervous system (ENS) [15,16,17].

More advanced therapies represented by deep brain stimulation, MRI-guided focused ultrasound, and therapy with levodopa-carbidopa enteral suspensions can support individuals with medication-resistant tremor and dyskinesias [18,19].

## 2. Gut–Brain Axis

The gut–brain axis is defined as the two-way communication among the central nervous system (CNS) and the ENS that bridges emotional and cognitive brain areas with outermost intestinal functions [20]. This communication comprises the CNS (brain and spinal cord), the autonomic nervous system, the ENS and the hypothalamic pituitary adrenal (HPA) axis [20].

Sympathetic and parasympathetic limbs of the autonomic system participate in afferent (vagus nerve to CNS) and efferent pathways (CNS to ENS) [21].

The hypothalamic pituitary adrenal gland is known to have roles in the adaptive responses to several stressors and it is a component of the limbic system that is crucial in emotional and memory processes [22]. Stressors together with elevated levels of pro-inflammatory cytokines induce the release of corticotropin-releasing factor (CRF) from the hypothalamus that activates this system. The release of adrenocorticotropic hormone (ACTH) from the pituitary gland leads to cortisol secretion from adrenal glands [23]. Cortisol is a crucial stress hormone that interacts with several organs, brain included. Therefore, activities of intestinal functional effector cells, such as immune cells, enteric neuronal cells, smooth muscle cells, interstitial cells of Cajal and enterochromaffin cells are affected by both neuronal lines and hormones [24]. Moreover, these cells are influenced by gut microbiota [25].

As mentioned above, GI dysfunction is a crucial non-motor symptom of PD, which frequently appears at the very early stage of the disorder. Some studies established that PD patients are affected by constipation for more than 20 years before the onset of motor symptoms [26]. GI symptoms, including exaggerated salivation, dysphagia, constipation, esophageal motility disorder, and gastric abnormalities, frequently occur years before motor symptoms and their incidence in a healthy population has been associated with an increased risk of developing PD [27]. Additionally, the association with GI abnormalities validates the Braak’s theory that PD might initiate in the GI tract, supported by the presence of Lewy body burden in the ENS with respect to other body areas and in the CNS [28].

Observing the distribution of Lewy bodies in PD patients, Braak and his research group assumed that α-synuclein pathology begins in the ENS and then travels to the brain passing through the brainstem, midbrain, basal forebrain and lastly the cortical regions [29,30]. α-synuclein aggregates are usually found in structures that contribute to the parasympathetic innervation to the intestine [31]. For this reason, the vagal nerve may represent the communication channel between the gut and brain. In fact, recently it has been proved that full truncal vagotomy can be related to a decreased risk in developing PD compared to highly selective vagotomy or control conditions [32,33]. Solid data are still lacking regarding this hypothesis, and several studies show contradicting evidence.

Interestingly, a clinical study showed that there is no significant association between *T. gondii* infection and idiopathic PD [34] and another work showed a significant increase in dopamine metabolism in neural cells [35]. Moreover, a very recent paper goes against Braak’s theory indicating that α-synuclein aggregates are found in the vagus nerve and in the stomach of PD patients, but not in normal ageing people, thus suggesting the beginning of α-synuclein pathology in the brain [36].

α-synuclein accumulation in the GI system is correlated with injury in the enteric neurons and feasibly triggers GI dysfunction [37]. The damage involves both myenteric and submucosal plexuses of the gut and the distribution is through the whole GI tract, starting from the esophagus to the rectum [38]. Accordingly, it has been proved that α-synuclein can be retrogradely moved from the intestine to the brain in rat models. Emerging studies in vitro and in vivo showed that α-synuclein can diffuse through endocytic mechanisms to neighboring neurons where it creates inclusions [38]. Another evidence showed that a PD-like pathology can be induced by oral administration of rotenone in mice, thus proving that the local effect of pesticides at the level of the ENS may be enough to induce PD signs from the ENS to the brain [39].

Since the olfactory bulbs and the ENS are continuously exposed to environmental agents throughout inhalation or ingestion, may be possible that factors such as diet, toxins, microorganisms and different environmental pathogens may determine and propagate PD pathology progression probably against a background of genetic susceptibility [40].

## 3. Gut Microbiota

The human gut is home to numerous bacteria, archaea, fungi, microbial eukaryotes and viruses/phages. This assortment of microbes is named the “gut microbiota” and their respective genes are the “microbiome” [41,42]. The typical composition of gut microbial community includes *Bacteroidetes*, *Firmicutes*, *Actinobacteria*, *Proteobacteria* and *Verrucomicrobia phyla*. *Bacteroidetes* and *Firmicutes*, in a healthy individual, represent more than the 90% of the total bacteria and their ratio depends on the host genomes and environmental features (hygiene, use of antibiotics or drugs, lifestyle and diet) [43]. Several pathologies affect the gut inducing dysbiosis. These pathologies include obesity, diabetes, diarrhea and irritable bowel syndrome [44].

The gut microbiota in adults is quite stable but the bacterial community can be easily altered by environmental factors. Dietary habits represent the main factor that influence the microbiome variety and the effects of nutrients and diet on gut microbiota were widely studied [45]. Emerging evidence shows that the interaction between diet and microbiota is highly dependent on the individual microbiota composition. 

Gut microbes contribute to the digestion, absorption, metabolism and transformation of undigested macronutrients into advantageous and active molecules for human health [46]. Every macronutrient impacts the gut microbiota in a distinctive manner, and changes in macronutrient proportions, quantities and categories are associated with microbiota composition [47]. Carbohydrates represent the most effective macronutrients for modifying the gut microbiota, in particular dietary fiber [48]. 

Nowadays the identification and quantification of gut bacterial genera has enabled investigation of the impact of diet on gut microbiota. In a dietary intervention period ranging from 24 to 48 h it has been shown that quick variations occur in microbial composition at species and family levels (but not *phyla*) [49]. Similarly, rodent models indicate that changes in macronutrient intake can modify gut microbiota composition in a day [50,51,52]. These reported changes were caused not only by the diet composition, but also by intrinsic and extrinsic factors that may play a role, such as circadian rhythm and feeding behaviors [53,54]. Emerging evidence shows how specific bacteria respond to certain dietary components. Protein, fats, digestible and non-digestible carbohydrates, probiotics and polyphenols all produce changes in the microbiome with secondary consequences for host immunologic and metabolic markers. 

The impact of dietary proteins on the gut microbiota was first reported in 1977. A culture-based study showed low levels of *Bifidobacterium adolescentis* and an improved number of *Bacteroides* and *Clostridia* in humans with a high beef diet compared to subjects with a meatless diet [55]. In vegetarians, it has been reported that the intake of whey and pea protein extracts increases gut-commensal *Bifidobacterium* and *Lactobacillus*, moreover, whey protein reduces the pathogenic *Bacteroides fragilis* and *Clostridium perfringens* [56]. Pea protein intake was also associated with increased intestinal short-chain fatty acid (SCFA) levels, beneficial for the maintenance of the intestinal barrier [57]. Conversely, the number of bile-tolerant anaerobes including *Bacteroides*, *Alistipes* and *Bilophila* is associated with animal-based protein intake together with a reduction in the number of the *Roseburia E. rectale* group [52,58,59]. 

Dietary polyphenols, such as flavonoids, anthocyanins and phenolic acids are widely investigated because of their antioxidant properties. These compounds are contained in many foods like fruits, seeds, vegetables, tea, cocoa products and wine [60] and they usually determine the enrichment of *Bifidobacterium* and *Lactobacillus genera* in the gut [61,62,63,64,65]. *Bifidobacterium genera* are known to have several health benefits included immune-modulation, cancer prevention and inflammatory bowel disease management [66]. Interestingly, an investigation of the antibacterial activity of fruit polyphenols demonstrated that enteropathogens such as *Staphylococcus aureus* and *Salmonella typhimurium* are particularly susceptible to these polyphenols [67]. Moreover, low levels of infective *Clostridium* species (*C. perfringens* and *C. histolyticum*) after fruit, seed, wine and tea intake were observed [62,65,68,69].

The standard Western diet is rich in saturated and trans fats but low in mono and polyunsaturated fats. A high saturated fat intake has been shown to increase the number of total anaerobic microflora and the relative quantity of *Bacteroides* and *Bilophila* [51,70]. There are no clinical studies demonstrating an alteration of gut microbiota upon high-unsaturated fat diets; however, in vivo studies (on mice) reported an increase in in Actinobacteria (*Bifidobacterium* and *Adlercreutzia*), lactic acid bacteria (*Lactobacillus* and *Streptococcus*), and Verrucomicrobia (*Akkermansia muciniphila*) [71].

Alcohol consumption could also modulate gut microbiota. Alcohol absorption occurs predominantly in the small intestine [72] and it is mostly metabolized in the liver by alcohol dehydrogenase (ADH), which converts alcohol into acetaldehyde which is lethal for tissues and gut microbes. Moreover, alcohol consumption could lead to a disruption in the microbiota homeostasis, increasing the number of gram-negative bacteria [73,74], decreasing the number of SCFA-producing bacteria [75], damaging intestinal barrier integrity through toxins produced by gram-negative bacteria [76] and increasing the permeability of the intestinal mucosa [77].

Increasing evidence proposed that alcohol intake may directly affect the gut microbiota composition. Alcohol consumption in rats for 13 weeks led to reduced α-diversity and β-diversity, reduced abundance of *Lactobacilli*, and enhanced *Bacteroidetes* compared to the control group [72].

## 4. Gut Microbiota and PD

Evidence from Yang et al. [78] revealed that microbiota dysbiosis could represent a cause of PD. Specifically, in a mouse model treated with rotenone, modifications of fecal bacterial compositions displayed by a decrease in bacterial diversity and high levels of *Firmicutes* and *Bacteroidetes* preceded the instauration of α-synuclein pathology [78]. In humans, gut microbiota composition varies as the PD progresses and these alterations are correlated with PD clinical symptoms [79]. Even though gut microbiota dysbiosis in PD are well known, it is still not clear whether alterations in the intestinal microbiota are either a cause or an effect of the disease.

Nowadays, it is well-established that the microbiota composition is altered in PD patients [80,81,82,83,84]. The first evidence is from 2015 [83], in particular, fecal samples of PD patients presented an important reduction in the number of *Prevotellaceae* (77.6%) together with an increase in the number of *Enterobacteriaceae*, compared with healthy population, that result correlated with the severity of postural instability and gait trouble [85]. The reduction in the number of *Prevotellaceae* diminishes the levels of beneficial neuroactive SCFAs and decrease the biosynthesis of thiamine and folate, thus suggesting a cause of the lack of these vitamins in PD patients [86,87].

A decrease in *Prevotellaceae* in PD patients may be related to the reduction in mucin production defining an amplified intestinal permeability, known also as “leaky gut”, that can be associated with α-synuclein aggregates through the bacteria translocation and the production of bacterial products that induce inflammation [88] (i.e., gut-derived lipopolysaccharide (LPS) that can support the impairment of the blood–brain barrier [89]) and the reactive oxygen species production in the GI system thus starting the α-synuclein accumulation at the level of ENS [90]. In an in vivo study, using rats injected at the level of *substantia nigra* with LPS, an induction of inflammation was observed, leading to damage of the nigrostriatal dopaminergic neurons, thus suggesting that this event could be implicated in neurodegeneration processes [91]. Moreover, Gorecki et al. [92] further investigated the effects of LPS in the communication between gut microbiota and α-synuclein accumulation, overexpressing the gene of human α-synuclein in mice. Firstly, authors showed that the levels of mucin-degrading *Verrucomicrobiae* and LPS-producing *Gammaproteobacteria* were higher in fecal samples of severe PD patients, while in mice overexpressing the human α-synuclein, the number of *Verrucomicrobiae* was lower. Secondly, the researchers demonstrated that LPS exposure can alter the intestinal barrier function targeting cell membrane tight junctions. Moreover, LPS intake in in vivo studies using an α-synuclein-overexpressing mouse model showed the manifestation of early motor impairment, thus supporting the hypothesis of proinflammatory gut microbiome environment as a leading cause for PD pathogenesis [92].

Recently, it has been demonstrated that gut microbiota transplants from parkinsonian mice into normal mice C57BL/6 was found to be associated with motor impairment and striatal neurotransmitter reduction. Specifically, sequencing of 16S rRNA showed that phylum *Firmicutes* and order *Clostridiales* diminished, while phylum *Proteobacteria*, *Turicibacterales* and *Enterobacteriales* were higher in fecal samples of parkinsonian mice, in parallel with enhanced fecal SCFAs. Notably, the fecal microbiota transplant (FMT) in MPTP-induced PD mice exerted neuroprotective effects, indeed, it was able to inhibit the activation of microglia and astrocytes in the *substantia nigra*, counteracting gut microbiome alterations, decreasing fecal SCFAs, alleviating physical impairment and increasing striatal dopamine and serotonin release [93].

Phage-related dysbiosis in PD is known, but recent findings suggest that phages can represent a leading cause of α-synuclein misfolding and that lytic bacteriophages could have a pivotal role in PD onset [94]. For example, the lytic *Lactococcus* phages are more numerous in PD patients than in healthy people, which is related to a sharp decrease in *Lactococcus* bacteria [94] since these bacteria are able to release the neurotransmitter dopamine [95] and regulate gut permeability [96]. In fact, low levels of *Lactococcus* bacteria, triggered by the high amounts of strictly lytic phages in PD patients, might be involved in the generation of α-synuclein misfolding [29,97,98].

A recent study demonstrated that oral administration of microbial metabolites in germ-free mice may lead to neuroinflammation causing the development of motor function alterations [99]. In particular, it is well-known that the gram-negative bacterium *Helicobacter pylori* is a leading cause of various GI problems, mainly peptic ulcers [100]. Moreover, many studies demonstrated a causal link between *Helicobacter pylori* and PD [101,102]. Numerous mechanisms were proposed to link *Helicobacter pylori* with PD pathogenesis: it could release toxins that affect the CNS or through glycosylation, generating cholesteryl glucosides with the same molecular structure of toxin cycads. These cholesteryl glucosides are neurotoxic and can go across the blood–brain barrier (BBB), leading to dopaminergic neurodegeneration [103,104]. Furthermore, *Helicobacter pylori* can activate the immune system through the activation of immune mediators, such as monocytes and determining the release of both interleukins and cytokines that may cause an important neuroinflammatory response [105]. In line with this hypothesis, biopsies of colonic tissue form PD patients were analyzed, and it was revealed that in PD conditions there is an increased expression of pro-inflammatory cytokines, such as TNF-α, IFN-γ, IL-6 and IL1-β as well as the activation of enteric glial cells [106,107]. Neuroinflammation represents the leading cause of the disruption of the BBB, microglia activation and neurodegeneration and the presence of *Helicobacter pylori* may induce the production of autoantibodies against dopaminergic neurons that extend neuro-inflammatory processes [108]. Lastly, *Helicobacter pylori* is able to trigger apoptosis through the nitric oxide and mitochondrial Fas–FasL pathways that could lead to neurodegeneration [109].

An healthy microbiota composition is beneficial for competitive exclusion activities, immunological regulation and the production of substances such as vitamins, secondary bile acids and SCFAs [110]. Dietary fiber is used as a food source by a large number of colonic bacteria for the generation of metabolic bioproducts: in particular SCFAs [111].

Clinical studies reported a causal link between the decreased number of SCFA-producing bacteria (from the genera *Blautia, Coprococcus* and *Roseburia*) which contributes to the “leaky gut”, and the increased number of opportunistic pathogens and carbohydrate-metabolizing probiotics [112,113].

Moreover, a substantial decrease in acetate, propionate and butyrate in PD fecal samples compared with healthy controls has been observed [114]. This reduction in SCFA might have a crucial role in ENS alterations and gut dysmotility in PD. Additionally, the decrease in the number of *Prevotellaceae* and an increase in the number of *Lactobacilliceae* have been associated with lower concentrations of ghrelin in PD patients [115]. Ghrelin is a hormone produced by the gut involved in the homeostasis of the nigrostriatal dopamine function and PD patients show an impairment in ghrelin secretion [116,117].

Notably, it has been shown that carbohydrates themselves induce dopamine production from the brain by promoting the passage of tyrosine (dopamine precursor) across the BBB into cerebrospinal fluid [118]. Overall, a balanced diet of carbohydrate and protein mixture could ameliorate motor signs in PD patients [119].

Furthermore, many studies associated celiac disease, a gluten-induced GI disorder, with PD pathogenesis [120]. However, additional investigation must be conducted to clarify this association and the relevance of diet in PD.

Overall, the evidence reported suggests that gut microbiota is deeply altered in PD, as reported in clinical studies and in vivo models, and the normalization of this dysbiosis would open new therapeutic opportunities for this disorder, such as the use of nutraceutical approaches, including probiotics, prebiotics or synbiotics and microbiota transplantation approaches [115].

## 5. Diet and Gut Microbiota–Brain Axis in PD

Numerous epidemiological studies reported that diet affects (positively or negatively) the onset of neurodegenerative disorders, including PD. The PD microbiome is characterized by reduced production of SCFA and improved LPS and these alterations may promote the onset or exacerbation of PD [121]. As discussed above, diet strongly influences gut microbial composition, and the Western diet is correlated with enhanced risk for PD, while the Mediterranean diet (with high intake of dietary fiber [122])might be able to diminish PD risk [123]. 

In particular, studies on PD patients correlate total caloric intake of macronutrients and micronutrients with symptom severity, with greater caloric consumption related to worse symptoms [124]. Diets rich in animal saturated fat have been related to a higher risk of developing PD [125]. Other foods correlated with PD exacerbation include canned fruits and vegetables, soda, fried foods, processed food, ice cream and cheese (all typical of the Western diet). Mechanistically, this may be due to the high amount of LPS-containing bacteria in the intestinal microbiome which affects gut barrier integrity, leading to endotoxemia (i.e., systemic LPS), NLRP3 inflammasome activation, insulin resistance and mitochondrial impairment and gluconeogenesis [123]. Conversely, a “healthy” diet increased the number of SCFA-producing bacteria and induced the release of components including glucagon-like peptide 1 (GLP-1) and brain-derived neurotrophic factor (BDNF), positively influencing PD risk [126].

The Mediterranean diet is related to a lower risk of PD onset [127]. Specific components of the Mediterranean diet are the reason of this positive effect, including fresh fruits and vegetables, nuts and other dried fruits, olive oil, wine and spices. Specifically, consumption of flavonoid-rich foods (berries, fruits, tea and wine) positively affects the risk of developing neurodegenerative disorders, including PD [128]. Polyunsaturated fatty acids (PUFA) are also inversely related to PD risk (elevated intake of ω3 fatty acids is linked to decreased PD risk) supporting the effects of fat consumption on the brain [129,130].

A daily diet enriched with plant carbohydrates and fiber is able to increase some particular macronutrients that PD patients lack. In contrast, a Western diet rich in refined carbohydrates and saturated fats, high fat goods and whole dairy products could lead to gut dysbiosis and may be implicated in PD pathogenesis [131,132]. Furthermore, antibiotics and microbial toxins produced by gut bacteria, comprising LPS and epoxomicin, may determine substantial variations in the gut microbiota and inflammation [133,134]. Several studies support the idea that differences in lifestyle are implicated in PD pathology. It has been proved that coffee and smoking may contribute to reducing the development of PD and this effect could be mediated by the gut microbiota. Beneficial effects of coffee and smoking could be due to the role of gut microbiota in mitigating intestinal inflammatory mechanisms [135]. Further studies showed that also red wine and tea may counteract PD predisposition [136].

While there are no therapeutic approaches that can avoid or delay PD by directly targeting the gut microbiota–brain axis, diet may influence both the gut microbiota–brain axis by modifying the microbiota composition and the neuronal functions of the ENS and CNS to ameliorate the progression of PD pathogenesis [137]. Recent investigations have shown that specific nutritional membrane precursors and cofactors are able to improve synaptic loss and membrane-related ENS and CNS impairments in PD and reduce motor and non-motor signs in preclinical studies [138,139]. The combination with the intake of prebiotic fiber may determine an amelioration in the treatment effects [126]. Moreover, oral administration of two circulating phosphatide precursors (uridine, and docosahexaenoic acid) was linked with an amelioration in dopaminergic neurotransmission, synaptic membrane formation and the density of dendritic spines [140,141,142].

## 6. Probiotics Interventions in PD

Numerous studies have shown that specific probiotics mixtures are able to restore gut microbiota and improve immune response [143]. Probiotics are live microorganisms that when administered in sufficient amounts can promote a restoration of gut microbiota and ameliorate immune homeostasis in the host [144].

The most commonly used probiotic bacteria are *Lactobacilli*, *Enterococci*, *Bifidobacteria*, yeasts and combinations of different beneficial bacteria [145]. Therapeutic and prophylactic effects exerted by probiotics intake are thought to be mediated throughout a broad range of mechanisms. Gut microbiota can be affected by probiotic supplement through competition with nutrients, adhesion to the intestinal epithelium, antagonism and cross-feeding [146]. Three-dimensional bacterial communities surrounded by self-produced extracellular matrices by probiotic bacteria (Biofilms) stimulate the colonization and extended duration in the GI system of the host and prevent the mucosal enrichment of pathogenic bacteria [147].

This process is caused by the release of organic acids (i.e., lactic acid by *Lactobacillus* and *Bifidobacterium* species) that lower GI pH, and bacteriocins that together can counteract pathogens’ proliferation in the human GI system and urinary system [148]. Notably, treatment with *Lactobacillus* probiotics (in particular *Lactobacillus casei*) during *Helicobacter pylori* eradication therapy, ameliorated eradication efficacy, apparently through their antagonistic mechanisms against *H. pylori* [149]. Cross-feeding between probiotic bacteria and host microbiota can support the production of SCFAs such as butyrate in the gut [150].

Probiotics can also modulate a wide range of host immune functions that include both innate and adaptive (both cell-mediated and humoral) immunity. Particularly, probiotics are able to improve phagocytosis and enhance secretion of antibodies, generating increased immunological defenses against pathogens [151]. Additionally, probiotics can release a wide range of anti-inflammatory factors, downregulating pro-inflammatory cytokines [152], potentially counteracting intestinal inflammation. Moreover, probiotics can ameliorate GI barrier function [153]. For example, *Lactobacillus* and *Bifidobacterium* species can overexpress tight junction proteins and stimulate mucus secretion that can avoid the adhesion of detrimental microorganisms [154]. A growing body of evidence has demonstrated the protective effects exerted by probiotics in ameliorating intestinal epithelial integrity, counteracting barrier disruption, promoting healthy homeostasis of the mucosal immune system and blocking pathogenic bacterial proliferation [155,156].

Additionally, several strains of probiotic bacteria can stimulate intestinal motility and reduce GI alterations as demonstrated by a study in aging patients, where probiotics were able to modify bowel movements, reducing symptoms such as diarrhea and constipation [157]. Moreover, probiotics have a role in alleviating symptoms connected with lactose maldigestion due to the presence of enzymes such as β-galactosidase and bile salt hydrolase, which ameliorate lactose digestion in the host system [158].

Notably, many studies demonstrated that it is possible to regulate brain functions by ameliorating anxiety and depression with probiotic supplement. In an in vivo model of autism spectrum disorder (ASD), Hsiao et al., [149] indicated that *Bacteroides fragilis* administration counteracted the alterations in gut permeability and ameliorated ASD symptoms [159]. Further studies on animals showed that probiotic administration (i.e., *Lactobacillus plantarum*, *L. rhamnosus*, *B. longum*) can have anti-anxiolytic and antidepressant effects and modify cognitive activity [160].

In fact, it has been shown that probiotics can release a wide variety of bioactive compounds that can impact the host and its microbiota. Particularly they can release neuroactive compounds such as oxytocin, gamma-aminobutyric acid (GABA), serotonin, tryptophan, tryptamine, noradrenaline, dopamine, and acetylcholine [161].

Intake of specific probiotics also showed positive consequences for brain performances in clinical studies. The ingestion of *Lactobacillus casei* strain *Shirota* in chronic fatigue syndrome patients could drastically counteract anxiety [162].

Studies on the administration of probiotics for PD treatment are very limited. One study reported that PD patients with chronic constipation taking fermented milk containing *Lactobacillus casei Shirota* for five weeks ameliorated fecal consistency and diminished bloating and abdominal pain [163]. Although probiotics can represent a valuable tool to counteract alterations in PD microbiota composition and ameliorate GI function by reducing gut leakiness, bacterial translocation and related inflammation in the ENS, ameliorating GI functions with probiotics might not only improve GI function and/or protection of the GI system but also increase levodopa absorption and counteract motor and cognitive impairment including anxiety, depression and memory difficulties [164,165], which are common symptoms in PD patients.

The most commonly used probiotics such as *Lactobacilli*, *Enterococci*, *Bifidobacteria*, yeasts and specific mixtures [144,145] may modulate brain function by ameliorating anxiety and depression [162]. In fact, in clinical studies and in vivo models of PD, probiotics were able to alter the composition of gut microbiota and consequently may improve gastrointestinal function, neuroinflammation and even levodopa absorption [164].

Several in vivo and in vitro models were considered to study the neuroprotective effects of probiotics and their use as a potential treatment for PD [166]. In particular, mouse models are the most commonly used. Hsieh and his research group [167] compared the motor functions upon probiotic administration vs. vehicle in a MitoPark PD mouse model. The probiotic mixture was composed of six common probiotic strains (*Bifidobacterium bifidum*, *Bifidobacterium longum*, *Lactobacillus rhamnosus*, *L. rhamnosus GG*, *Lactobacillus plantarum LP28* and *Lactococcus lactis* subsp. *Lactis*) and they observed an amelioration in motor performances of mice treated with the probiotic. In particular, mice showed better gait, balance and coordination from the 16th week after supplementation. Additionally, upon the treatment, they displayed decreased loss of dopaminergic neurons, thus suggesting a neuroprotective effect of the probiotics [167].

Similarly, neuroprotective effects of another probiotic combination containing *L. rhamnosus GG*, *Bifidobacterium animalis lactis*, and *Lactobacillus acidophilus* were observed in in 1-methyl-4-phenyl-1,2,3,6-tetrahydropyridine (MPTP)- and rotenone toxin-induced PD mouse models [168]. In these models, the probiotic supplementation promoted the butyrate production, which plays a role in rescuing nigral dopaminergic neurons from MPTP- and rotenone-induced neurotoxicity. Moreover, high levels of BDNF and glial cell line-derived neurotrophic factor (GDNF) together with the inhibition of MAO-B were detected, which can lead to increased dopamine synthesis and the promotion of dopaminergic neurons survival thus helping cell survival and cell proliferation [168].

In another study, in 6-hydroxydopamine (6-OHDA) mice, a novel probiotic mix SLAB51 (sold as Sivomixx, composed of nine bacterial strains: *Streptococcus thermophilus*, *B. longum*, *Bifidobacterium breve*, *Bifidobacterium infantis*, *L. acidophilus*, *L. plantarum*, *Lactobacillus paracasei*, *Lactobacillus delbrueckii* subsp. *Bulgaricus* and *Lactobacillus. brevis*) was administered [169]. Notably, this formulation was associated with neuroprotection with a reduction in dopaminergic neuronal loss in the substantia nigra and striatum, assumed to be mediated through the activation of the peroxisome proliferator-activated receptor gamma (PPAR-γ) by microbial metabolites, thus leading to anti-inflammatory and antioxidant effects, as well as an increase in BDNF and consequently the activation of its pro-survival pathway [169].

Moreover, probiotics can be genetically manipulated to increase their beneficial effects. In a recent study, Fang et al. used *Lactococcus lactis cremori* carrying a GLP-1 expression vector as a treatment for an MPTP mice model [170]. Treated mice showed increased expression of tyrosine hydroxylase into the nigrostriatal pathway, reduced locomotor impairment and lower inflammation, compared with the control group. Moreover, the probiotic was also able to counteract the proliferation of intestinal pathogen Enterobacteriaceae, increasing the number of probiotic *Lactobacillus* and *Akkermansia* species. Interestingly, GLP-1 can cross the BBB and binds GLP-1 receptors in the brain. This insulin-signaling pathway is crucial in neurogenesis, neuronal metabolism and synaptic plasticity [170], and GLP-1 agonists are currently treatments of clinical trials on PD patients [171]. 

In a *Caenorhabditis elegans* α-synuclein model of PD, treatment with *Bacillus subtilis* PXN21 was associated with a reduction in α-synuclein accumulation in the host [172]. *Bacillus subtilis* PXN21 could exert neuroprotective effects through modifications of host sphingolipid metabolism. This results were in line with the hypothesis that an alteration in lipid metabolism, in particular ceramides and sphingolipids, contributes to PD pathogenesis [173]. Additionally, the beneficial effect of *B. subtilis* is partially due to a biofilm formation in the gut of the model [172]. Furthermore, an in vitro study demonstrated that, by co-culturing peripheral blood mononuclear cells isolated from PD patients with probiotic species (*Lactobacillus salivarius*, *L. plantarum*, *L. acidophilus*, *L. rhamnosus*, *Bifidobacterium animalis* subsp. *lactis* and *B. breve*), the release of pro-inflammatory cytokines was inhibited, in parallel with stimulation of the release of anti-inflammatory cytokines [174]. Among the tested probiotics, *L. salivarius* and *L. acidophilus* showed leading activities. Moreover, in this study, the probiotics were able to inhibit the proliferation of potentially pathogenic bacteria such as *Escherichia coli* and *Klebsiella pneumoniae* [174]. 

## 7. Prebiotics Intervention in PD

Also, prebiotics represent non-digestible compounds that may be beneficial for the host by modulating the gut microbiota [175]. A prebiotic is defined as “a substrate that is selectively used by host microorganisms exerting a health benefit”. Most of the prebiotic are fermentable dietary fibers but not all the dietary fibers are prebiotics. Commonly, the consumption of a high percentage of fiber in the diet promotes an in increase in bacterial diversity and leads to an expansion and/or an increase in the activity of beneficial bacteria (i.e., *Bifidobacterium* sp., *Lactobacillus* sp., *Akkermansia* sp., *Faecalibacterium* sp., *Roseburia* sp., *Bacteroides* sp. and *Prevotella*) together with a decrease in the number of detrimental bacteria (e.g., *Enterobacteriaceae*) [176].

Examples of prebiotics include pectins, inulin, fructo-oligosaccharides (FOS) and galacto-oligosaccharides (GOS). Particularly, the chemical characteristics of fibers, such as polymerization, solubility and viscosity determine the metabolism inside the GI tract, resulting in definite microbiota transformations after the ingestion [177].

Preclinical and clinical studies showed that the intake of wholegrain food that contains β-glucans (soluble non-starch polysaccharides) helps the growth of *Lactobacilli* and *Bifidobacteria* in rats and humans. It has been reported that supplementation with intact cereal fibers (i.e., wholegrain cereals, barley fibers, wheat bran and rye fibers) supported the growth of *Actinobacteria*, *Bifidobacterium, Clostridium, Lachnospira, Akkermansia,* and *Roseburia* in humans. Finally, the consumption of resistant starch determined the proliferation of *Bifidobacterium, Faecalibacterium* and *Eubacterium*, while reducing the amount of *Ruminococcus* strains [178,179].

Moreover, fiber solubility also has an impact on the gut microbiome. Soluble fiber has a stronger effect on microbial composition and diversity in piglets in comparison with insoluble fiber. However, cellulose—an insoluble and non-fermentable fiber and a source of fiber in fruit and vegetables—is transformed by *Ruminococcus* and *Fibrobacter*, which for this reason are called “cellulose-degrading microbes” [180]. In vivo studies have shown that cellulose intake leads to an increase in microbial species such as *Eptostreptococcaceae, Clostridiaceae*, *Akkermansia*, *Parabacteroides*, *Lactobacillus*, *Clostridium*, *Eisenbergiella*, *Marvinbryantia*, *Romboutsia*, *Helicobacter*, *Enterococcus* and *Desulfovibrio* together with a lower proliferation of *Sutterellaceae, Lactobacillaceae* and *Coriobacteriaceae* [43].

In addition to the effects on microbiota composition, dietary fibers have a role in microbial enzymatic function and in metabolite absorption. Chemical properties, including fiber solubility and fermentability influence the degree and location of microbial fermentation and which type of metabolite is produced. Two important fibers are GOS, based on lactose and FOS, synthesized from fructose [181]. GOS and FOS arrive to the colon mainly unaltered and are mostly transformed by *Bifidobacteria*. Metabolic products such as SCFA, lactose, hydrogen, methane and carbon dioxide induce an acidic milieu in the colon, which leads to death, or reduced multiplication, of deleterious bacteria [182].

A large number of clinical studies demonstrated that lowering carbohydrate intake or wholegrain cereals reduced the amount of butyrate-producing bacteria, such as *Bifidobacteria*, as well as SCFAs themselves [183]. Soluble and fermentable fiber can intensify the microbiota enzymatic activity to transform complex carbohydrates in health-promoting SCFAs such as acetate, propionate and butyrate. These SCFAs, in particular butyrate, have been involved in colonocyte metabolism, thus helping intestinal barrier functionality, in glucose homeostasis, lipid oxidation and they have anti-inflammatory and mucosal immunomodulatory effects [177]. 

While insoluble fibers (i.e., cellulose) are not implicated in SCFA production, it has been observed that they probably have a role in the linoleic acid, nicotinate and nicotinamide, glycerophospholipid, glutathione and sphingolipid pathways as well as the valine, leucine and isoleucine metabolic pathways [184].

In fact, through computational-experimental framework a relationship between PD and branched-chain amino acid transferase 1 (BCAT-1) was found. This enzyme is involved in the first step of branched chain amino acid (BCAA) catabolism [185] and it has been discovered that BCAT-1 levels are usually high in PD-susceptible regions of the healthy human brain, and that its expression is lower in the substantia nigra of sporadic parkinsonian patients [186]. While this correlation has been observed, further studies on in vivo models are needed to clarify the underlying mechanisms. Moreover, glutathione exerts anti-oxidant effects reducing reactive oxygen species (ROS) [187] and glutathione S-Transferases (GSTs) enzymes catalyze the conjugation of glutathione to various electrophiles and the role of GSTs in the protection of dopaminergic neurons has been examined using several models of Parkinson’s disease [188].

Prebiotic fibers may be beneficial in the activity of the immune system, bowel mobility and constipation. For these reasons, enriching the diet with prebiotics might be beneficial for inflammation and GI alterations occurring in PD patients. Furthermore, it has been observed that GOS and FOS determine an increase in BDNF levels at the level of the dentate gyrus of the hippocampus in rats [189]. Since BDNF is a neurotrophin implicated in neuronal protection, survival and plasticity, GOS and FOS supplementation in the diet might affect brain health. Despite all the evidence reported, the use of prebiotics in patients with PD has not yet been investigated but since PD patients present a lower abundance of SCFA butyrate-producing bacteria, prebiotic fibers may be used as a supplement to correct this dysbiosis [190,191]. Notably, SCFAs are able to activate microglia, inducing T-regulatory cells to increase cytokine release to regulate neuroinflammatory mechanisms [99].

## 8. Synbiotic Intervention in PD

The concept of synbiotic indicates food components or dietary supplements fusing probiotics and prebiotics [192]. In particular, synbiotics arise from the necessity to overcome possible probiotics survival difficulties, for this reason in these formulations the prebiotic compound must selectively promote the activity and the survival during the passage through the upper GI tract of the probiotic fraction [193]. Synbiotics are beneficial by favoring the survival and implantation of microbial supplement in the GI system. In fact, they selectively induce the proliferation and activate the metabolism of a small group of healthy bacteria. A wide range of factors such as pH, hydrogen peroxide_,_ organic acids, oxygen and moisture stress affect probiotic viability [194]. Most commonly used probiotic strains in synbiotic formulations include *Lactobacilli*, *Bifidobacteria* spp., *S. boulardii* and *B. coagulans*, while the prebiotics used include oligosaccharides such as FOS, GOS and xyloseoligosaccharide (XOS), inulin and prebiotics from food like chicory and yacon roots. [66].

The positive effects exerted by synbiotic intake in clinical investigations comprise: (1) balanced gut microbiota increasing the levels of *Lactobacilli* and *Bifidobacteria,* (2) amelioration of liver function in cirrhotic patients, (3) enhancement of immune system function and (4) inhibition of bacterial translocation and decreased occurrence of nosocomial infections in patients after surgery [195].

Constipation is one of the main symptoms in PD patients affecting their quality of life. Synbiotics may be useful for PD-related non-motor side effects by ameliorating immune function, dysbiosis and bowel functions. In a clinical study, *Lactobacillus salivarius* was able to decrease inflammatory markers in healthy subjects with a higher effect in combination with prebiotics [196]. In a different investigation, treatment with synbiotic yogurt containing *Bifidobacterium animalis* and prebiotics in females with constipation, produced increased gut movement and defecation, compared to controls [197]. Moreover, it has been assessed that the daily intake of a fermented milk containing multiple probiotic strains and prebiotic fiber for four weeks was able to increase in the number of complete bowel movements in patients with PD [124].

Another main symptom of PD is small intestinal bacterial overgrowth (SIBO) and patients that are SIBO positive usually present increased motor dysfunction [198]. Khalighi et al., [199] demonstrated that the association between antibiotic treatment and synbiotic supplementation containing *Bacillus coagulans* and prebiotic ameliorated the treatment response. Moreover, it decreased abdominal pain, flatulence and diarrhea [199].

Overall, the evidence reported in this review support the potential of probiotic, prebiotic and synbiotic supplementation in PD patients.

## 9. Discussion and Conclusions

PD is a one of the most common neurodegenerative disorders, characterized by motor and non-motor sign and symptoms, including gut dysfunctions, which may appear before the motor symptoms. PD underlying mechanisms involve increased oxidative stress and neuroinflammation [200]. So far, the existing therapies can alleviate PD-associated symptoms, but there is no cure to control the development and exacerbation of this disorder. Accumulating evidence suggests a crucial role of gut microbiota and an influence on the CNS, via the gut–brain axis, mediating different pathways. In particular, a healthy microbiota is correlated with lower risk of developing CNS disorders, including PD, while microbiota dysbiosis is correlated with higher incidence of PD. Diet may influence both positively and negatively the development of neurodegenerative disorders. Specifically, the Mediterranean diet (rich in fibers, flavonoids and PUFA) has positive effects on the gut microbiome and thus may reduce the development or exacerbation of PD; on the other hand, the Western diet (rich in meat, processed food and fried food) could lead to detrimental effects on the gut–brain axis (Figure 1). 

Numerous preclinical and clinical studies suggested that dietary interventions with prebiotics, probiotics or synbiotics, by modifying the microbiome composition, may improve brain health and decrease the risk of developing PD (Figure 1).

Dietary interventions are of high importance in particular at the very early stages of PD. Patients with PD may experience non-motor symptoms at early stages such as constipation, dysphagia, hyposmia and depression, that may influence dietary choices and thus may be responsible for the alterations of nutritional status reported in PD [201,202]. For example, as we mentioned above, PD patients are affected by constipation for more than 20 years before the onset of motor symptoms, therefore the use of nutraceutical interventions at this stage, including prebiotics, probiotics or synbiotics may be of high relevance [127,203].

Further studies in PD should take into consideration the role of the gut–brain axis and a deeper investigation into the underlying mechanism is required. Moreover, additional studies are needed to define the potential beneficial effects of the use of prebiotics, probiotic and synbiotics in maintaining protein and oxidative homeostasis in the ENS and to better understand the biochemical influences of these interventions on people affected by neurodegenerative diseases. Another point to be considered is whether constant exposure to probiotics, prebiotics or synbiotics affects the gut microbiome composition in a long-term way or, once the intervention is ceased, the microbiome composition would revert. 

Also, it is important to design studies considering the duration of intervention, the dosages and the combination of different interventions. Finally, in this scenario, it is crucial to define the best approach based on prebiotics, probiotics or synbiotics for PD and, analyzing the specific gut microbiome composition of a single patient, could help in creating a personalized therapy. Overall, we can postulate that prebiotics, probiotics and symbiotics may represent a potential therapeutic approach for PD.

## Figures and Tables

**Figure 1 nutrients-14-00380-f001:**
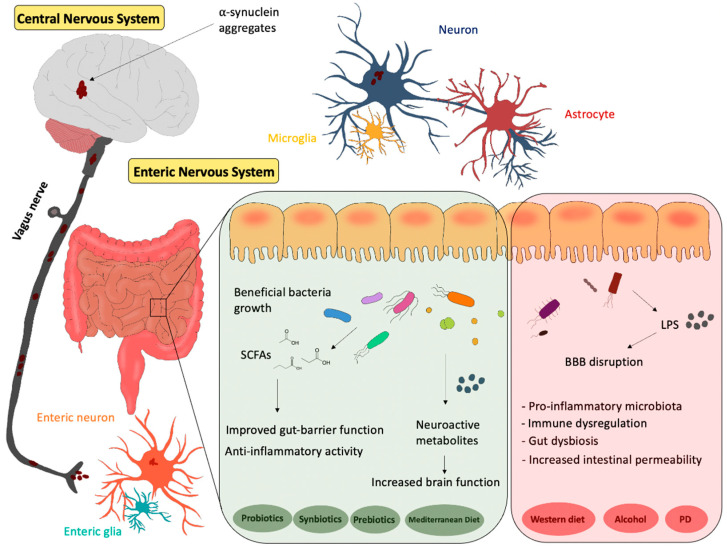
Representative scheme of the gut–brain axis. The influence of different diets and the potential effects of probiotics, synbiotics and prebiotics on gut microbiota. PD: Parkinson’s Disease.

## Data Availability

Not applicable.

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
