# Peer review of "Are We What We Eat? Impact of Diet on the Gut–Brain Axis in Parkinson’s Disease"

_nutrients, 2022, doi:10.3390/nu14020380_

Round 1

Reviewer 1 Report

The purpose of this review was to investigate the role of gut-brain axis and relevant dietary manipulations on PD. The manuscript is very well written and relative easy to be read by “Nutrients” audience. Only some minor points are needed to be clarified. Also, I have some suggestions to improve the review and manuscript in general. All these are comments in the pdf file.  

Author Response

Reviewer 1

The purpose of this review was to investigate the role of gut-brain axis and relevant dietary manipulations on PD. The manuscript is very well written and relative easy to be read by “Nutrients” audience.

RESPONSE: We would like to thank the Reviewer 1 for the positive comments and the time spent in reading our manuscript and for the comments provided that helped in improving our research article.

Only some minor points are needed to be clarified. Also, I have some suggestions to improve the review and manuscript in general. All these are comments in the pdf file.  

RESPONSE: We appreciate the Reviewer’s comments and we tried to address all the points raised. Please find the answers in the comment boxes.

Reviewer 2 Report

This draft review investigates the area of gut microbiota and dietary influence on gut brain axis with the PD as the prime example. I found it very thorough and comprehensive as it covers the risk factors around gut microbiota, and it also thoroughly discuss the potential dietary approaches for therapies/preventions.

 The draft is publishable in my view after addressing some minor issues as follows:

  • The title could be modified to reflect that focus of the review on PD.

  • In the abstract, I agree with all except the sentence “Gut microbiome restoration is able to avoid the PD progression and this effect could be executed by probiotics, prebiotics and synbiotics”. This is a strong statement that is not substantiated by the current evidence not from the review. Perhaps this sentence could be omitted or replaced by “Gut microbiome restoration through probiotics, prebiotics, synbiotics or other dietary means could have the potential to slow PD progression”.

Author Response

Reviewer 2

This draft review investigates the area of gut microbiota and dietary influence on gut brain axis with the PD as the prime example. I found it very thorough and comprehensive as it covers the risk factors around gut microbiota, and it also thoroughly discuss the potential dietary approaches for therapies/preventions.

RESPONSE: We would like to thank the Reviewer 2 for the positive comments and the time spent in reading our manuscript and for the comments provided that helped in improving our research article.

 The draft is publishable in my view after addressing some minor issues as follows:

  • The title could be modified to reflect that focus of the review on PD.

RESPONSE: We appreciate the Reviewer’s comment, we now modified the title as suggested.

  • In the abstract, I agree with all except the sentence “Gut microbiome restoration is able to avoid the PD progression and this effect could be executed by probiotics, prebiotics and synbiotics”. This is a strong statement that is not substantiated by the current evidence not from the review. Perhaps this sentence could be omitted or replaced by “Gut microbiome restoration through probiotics, prebiotics, synbiotics or other dietary means could have the potential to slow PD progression”.

RESPONSE: We totally agree with the Reviewer, and we replaced the sentence as suggested.
